# The Psychometric Characteristic of the Taekwondo Electronic Protector Cognition Scale: The Application of the Rasch Model

**DOI:** 10.3390/ijerph17103684

**Published:** 2020-05-23

**Authors:** Eun-Hyung Cho, Chang-Yong Jang, Yi-Sub Kwak, Eung-Joon Kim

**Affiliations:** 1Korea Institute of Sport Science, Seoul 01794, Korea; ehcho@kspo.or.kr (E.-H.C.); Jangcy529@kspo.or.kr (C.-Y.J.); 2Department of Physical Education, Dong-Eui University, Busan 47340, Korea; ysk2003@deu.ac.kr; 3Department of Physical Education, Korea National Sport University, Seoul 05541, Korea

**Keywords:** item response theory, electronic protector, taekwondo, scale, Rasch

## Abstract

This research was to investigate the psychometric characteristics of the electronic protector cognition scale by the infit and outfit of taekwondo athletes. Participants were 216 athletes (male = 109; female = 117) from 19 countries competed at the 19th Taekwondo World Championships. The electronic protector cognition scale consisting of 24-item with four subscales was utilized. The electronic protector cognition scale used a five-point Likert grading with 1 (not at all) to 5 (very likely). Analysis using IBM SPSS STATISTICS version 23 (IBM SPSS, Inc., Chicago, IL, USA) was conducted for the 226 data sets collected. WINSTEPS 3.74 (Linacre, 2015) was used for calculating subject reliability, item goodness-of-fit, scale propriety, and item level of difficulty, in order to apply the item response theory to the psychometric characteristics of electronic protectors. The research results showed that it was suitable for subject infit/outfit in taekwondo electronic protector cognition scale as 1.00~1.01 and the input/output of taekwondo electronic protector cognition scale as 1.00~1.01. Secondly, five-point scales were reviewed to be suitable for scale propriety, resulting from stage index judgment. Thirdly, 8 items showed problems in item goodness-of-fit. Finally, scale propriety was reported to be suitable considering the ability distribution of taekwondo players and the level of scale difficulty.

## 1. Introduction

The environment in sports, or tools, or equipment used in the game, has a direct or indirect influence on the results of an individual player or the team. These are important variables, having not only a direct but also an objective effect on the competition [1]. Taekwondo matches have undergone a multitude of changes, including the match environment and match rules. For instance, a full-body swimsuit in a swimming competition plays an important role in the performance, by increasing momentum and reducing low thrust, and the body and mind of athletes whose shoes change frequently in a marathon race. With the introduction of the electronic protector, significant changes are accomplished in the overall field and not just current partial changes. In particular, the athletic performance of athletes has been determined under the hypothesis that “the referee passes judgment with severity, and the best performer achieves the highest grade against the opponent, as whoever decides,” regardless of the characteristic of sports and the method of points assignment. It is a well-known fact that the referee’s decision ought to be uniform, irrespective of who is refereeing. As pointed out in various preceding researches, no steps have been taken to reduce the errors in refereeing, resulting in biased decisions of the referee. However, thorough training programs continue to be implemented in the operations of the various associations or leagues in order to improve the quality of the referee’s decision [2].

In order to discriminate between correct and incorrect refereeing, the electronic protector was developed and applied to the field as a mechanical means to determine factors, such as accuracy of strike intensity, consistency of scoring continuous blows, valid striking technique, and invalid batting skills [3]. The electronic protector has a cutting-edge electronic sensor attached to the body protector to determine the amount of impact upon hitting and automatically displays a point on the scoreboard. Various brands, such as ‘Daedo’ (Daedo International) and ‘LaJust’, apply the automatic recognition method, and KP&P uses both automatic and semi-automatic scoring methods interchangeably. The electronic protector was introduced at the Korea International Taekwondo Tournament in 2007, followed by the Beijing Olympics in 2008 and Taekwondo World Championships in 2009 [4]; all the matches at London Olympics 2012 were also refereed with the electronic protector. The media and various existing studies forecast positive aspects that can minimize the problem of the referee’s decision in taekwondo competition with the introduction of electronic protectors [5,6,7].

Conversely, a number of leaders and players who have experienced the electronic protector in the official games have been researched, making aware the negative aspects of the technology, namely, sensor malfunction in electronic arcs, plate arrest due to referee decision errors, sensor failure attached to the torso protector due to continuous competition, the problem of set sensor strength according to taekwondo player weight class, inaccuracy of major foot skills and strikes, incorrect marking of scores recognized for scoring, the problem of the size of the electronic catch, sensor and body protector technology attached to the foot sensor, and body protector due to introduction of electronic protector. All things considered, these results indicate that the electronic scoring system is as yet not standardized and is technically unstable, and reliability and propriety for its verdict are doubtful, even though the scoring system of the electronic body protector was introduced due to the concerns of the subjective nature of refereeing decisions [2].

A validity study is recognized as an important subject in the academic field of physical education. Thus, it is understood that there is a need for overall knowledge, such as the theoretical knowledge of measurement and the construct experience, the professionalism of the research plan, the analysis and interpretation of the data, etc. Moreover, the validity study is recognized as difficult because it requires continuous performance, not just a one-time occurrence. Thus, the validity study brings a focus to the integrated view and a single concept—the importance of construct validity is emphasized [8]. In this research, the Rasch model was employed in order to efficiently utilize the infit and outfit of the electronic protector cognition scale based upon critical problems regarding the score and verdict of the electronic protector in the existing study. On the one hand, the Rasch model is able to make up for the problem in the factorial experiment, i.e., the result can be different according to the characteristics of the population [9,10]. Conversely, the Rasch model is more adequate for providing the evidence of construct validity about how the individual item responds to the characteristic of the respondent, and whether a factor is recognized as suitable for the individual [11,12]. In addition, there is also an advantage of the Rasch model in the item goodness-of-fit, i.e., whether each question is suitable in the measurement of the psychological characteristics, in the item difficulty, i.e., how much the respondent agrees with the question, whether the Likert-scale functions as a measurement, and whether the model can evaluate the category propriety [13,14]. The characteristic of the measurement inspection constitutes the question of inspection and the theoretical index to understand the quality and the function of the inspecting tools [15,16,17]. The measurement characteristics to check the quality and function of inspecting tools for the personal evaluation are comprised of validity and reliability, which confirms the characteristics of the inspecting tools, level of difficulty and discriminant index, which checks the characteristic of the item, and inspection and item goodness-of-fit, which confirms the propriety of the inspection based upon the subject response [18]. It is meaningful to investigate the changes of the measurement characteristics for how taekwondo athletes on all continents perceive the electronic protector.

There are few existing studies for the measurement characteristics of taekwondo electronic protector cognition scales. Furthermore, the concrete research of multi-dimensional factors rather than that of a unidimensional factor is also a rare thing with regard to the problems of the electronic protector. Therefore, this research investigated the critical problems, such as fist scoring, intensity, sensor response and scoring part, wearing sensation (size), and how the electronic protector is perceived on the basis of the existing studies related to the score or verdict problems of the electronic protector. This research recognized the psychometric characteristics, applied them to scale propriety, and then investigated whether the infit and outfit is suitable using the inspection theory of item response theory.

## 2. Materials and Methods

### 2.1. Research Participant

This research was conducted by distributing questionnaires to players participating in the 19th Taekwondo World Championship (Copenhagen, Denmark, 14–18 October 2009; DAE DO INTERNATIONAL, COMPANY, BARCELONA, SPAIN) and the 90th National Sports Championship (Daejeon, South Korea, 20–26 October 2009; DAE DO INTERNATIONAL, COMPANY, BARCELONA, SPAIN), and randomly sampling about 250 participants of both sexes and all nationalities, The Taekwondo electronic protector was adopted. 

The competition was attended by 143 countries, 1011 athletes, 250 of whom answered the questionnaire, and 226 people used the questionnaire to analyze the data. Therefore, it was important to gather opinions from many players, not set standards, and the results presented included the country and the number of cases. The classification of continental athletes in Table 1 also shows how the gender and weight classes of the athletes in the questionnaire are distributed in Table 2. In other words, it would be good to see the results as a result of describing the general characteristics of the collected subjects. The weight category indicates that there is a difference between men and women (e.g., male pin weight: −54 kg, female pin weight: −47 kg).

The distribution and general features of respondents are specifically examined in Table 1 and Table 2.

### 2.2. Research Tools 

The questionnaire about the electronic protector cognition scale was to investigate the problems in the match field in order to measure how players perceived the electronic protector in taekwondo competitions. In the study of Jeon Ik-ki [3], the overall awareness of an electronic protector was performed by an in-depth analysis using the inductive content analysis and taxonomic analysis of the data. Based on classified contents, such as (1) intensity, (2) sensor response and scoring part, (3) fist score, and (4) wearing sensation, the research used these contents as measuring tools for data collection. The measurement was translated in English, and the taekwondo researchers and the professor for Measurement Evaluation in Physical Education confirmed the final translation after back translation. The measurement was classified in terms of 4 factors, such as intensity, sensor response and scoring part, fist score, and wearing sensation, i.e., 6 items of intensity, 8 items of sensor response and scoring part, 5 items of fist score, and 5 items of wearing sensation, making a total of 24 items. The scale was configured by a 5-point Likert scale from 1 (not at all) to 5 (very likely). Table 3 shows the main components of the indicators and contents of the questionnaire.

Both male and female competitors in the taekwondo competition were given a questionnaire of the 24 items regarding the electronic protector cognition scale, to verify the validity based on item response theory, for the purpose of this research. Therefore, this research did not analyze Cronbach’s alpha (α), i.e., construct validity and inner-item consistency, by the factor analysis of classic test theory.

### 2.3. Data Analysis

In this study, the data collected was analyzed using the IBM SPSS STATISTICS version 23 (IBM SPSS, Inc., Chicago, IL, USA). It was confirmed that the basic assumptions in item response theory were satisfied, after which the analysis was progressed. Frequency analysis was conducted initially to see the general characteristics of the research subjects. Next, using WINSTEPS version 3. 92 (SWREG, Inc. Shannon, Clare, Ireland), the subject reliability, item reliability, item number adequacy, item goodness-of-fit, item difficulty, the scale of difficulty to verify the validity of the scale were calculated. WINSTEPS 3.74 program, i.e., the analysis program applied to item response theory, was used to verify the item adequacy. The analysis by WINSTEPS was different from the analysis of the classic test theory in which the sum of scores was analyzed, and the overall trends were evaluated. This analysis meant that by using the analysis method of item response theory, each item was the research subject and a way of evaluating validity. The goodness-of-fit value (INFIT), i.e., the standard to determine lower validity items, is a criterion value 1.3, and the probability is less than 0.001 [19,20]. In addition, the average of the calculated mean square is 1.0; in other words, in cases of less than 0.75, it is evaluated as overfit or low mean square and misfit or high mean square [21]. In the case that the calculated goodness-of-fit value (infit, outfit) is less than 0.75 or more than 1.3, it is evaluated as a pointless question. Moreover, it is evaluated as to whether the phase conditioning value (Andrich Threshold’s α) gradually increases. If so, it is evaluated as appropriate to form a response category [22].

## 3. Results

### 3.1. Subject Reliability of the Scale

Table 4 shows the subject reliability in the taekwondo electronic protector cognition scale, where the infit 1.01 ± 0.2 and outfit 1.01 ± 0.2 were confirmed. This means that the subjects were consistently identified as appropriate, judging from the result of the infit and outfit analysis, i.e., within the range of 0.75–1.30. They were also trustworthy as the subject reliability was 0.88, and the model reliability was 0.90.

### 3.2. Item Reliability of the Scale

Table 5 shows the item reliability in the taekwondo electronic protector cognition scale. The infit and outfit of the taekwondo players showed infit 1.00 ± 0.19 and outfit 1.01 ± 0.19 values. The players were consistently identified as appropriate, judging from the result of the infit and outfit analysis, i.e., within 0.75–1.30. They were also trustworthy as the item reliability was 0.93, and the model item reliability was 0.94.

### 3.3. The Validity of the Scale Category

Table 6 shows whether the number of the taekwondo electronic cognition scale, i.e., the category number, is suitable. In this study, there were five categories in the taekwondo electronic cognition scale. All were valid since the categories, i.e., category label numbers 1, 2, 3, 4, and 5, were within the range of 0.75 to 1.30. In addition, the category of the five-point scale number was reported to be suitable because the coefficient appeared to increase step by step as the standard number increased in Andrich’s Threshold.

### 3.4. Item Goodness-of-Fit and Item Difficulty

Table 7 lists the calculated and arranged results of the item goodness-of-fit and item level of difficulty. In other words, it showed whether the taekwondo electronic cognition scale was suitable for the infit and outfit in terms of the level of difficulty. It was confirmed as valid since none of the 24 items deviated from the appropriate range of 0.75–1.30, from the result of item goodness-of-fit. On the one hand, item 6 (logit = 0.41), item 14 (logit = 32), item 9 (logit = 0.22) were easy to respond in the areas of intensity, wearing sensation, sensor response and scoring part, and fist score in the case of item difficulty. On the other hand, item 7 (logit = −0.62), item 1 (logit = −0.41), item 18 (logit = −0.35) were very difficult to respond in the areas of fist score and sensor response and scoring part.

Figure 1 shows whether the item difficulty is appropriate for the infit and outfit in terms of the level of difficulty. The distribution of respondents was on the left, while the position of items according to the capacity of item response was schematized on the right. As shown in the item goodness-of-fit, the group of respondents was mainly distributed with capacity 0 as the center. It confirmed that the capacity of respondents was distributed from about 3 to −4, while the level of item difficulty was distributed from −0.62 to 0.41. It is noticed that the level of item difficulty was distributed very narrowly. It meant that the capacity of respondents was evenly distributed, but there seemed to a problem of discerning the item itself effectively.

## 4. Discussion

The environment during sporting events or tools or equipment used in the game has a critical influence on the result of individual players or the team. In addition, they act as important variables with not only direct but also objective effects on the competition. In order to adjust to this change, the electric protector was developed and introduced in the taekwondo competitions. This research verified the validity of the measuring scale with the psychometric characteristics through the inspection theory of the item response theory, and about how critical problems, such as scoring intensity, sensor response, scoring part, wearing sensation (size), etc., were perceived by the competitors.

The main points found in this study were first, the results of taekwondo players’ response to four factors: strength, fist score, fit, sensor response, and score point for “electronic fixtures” showed that the reliability of the athletes themselves and confidence in the electronic fodder recognition scale were reliable. Second, the five-point Likert scale of the taekwondo electronic protector recognition scale, which consists of four factors (strength, fist score, wear, sensor reaction), was reasonable according to the Andrich threshold analysis. Third, as a result of the analysis of the conformity of questions in 24 questions of the electronic arc recognition scale, there were no problems within the statistical range of suitability, but as a result of the difficulty evaluation determined by the logit score, it was found that the level of difficulty recognized by the players was unique in eight questions. In conclusion, the recognition scale of taekwondo’s electronic protector was found to be a reasonable measure in the ability distribution and question-and-answer drawings of athletes, except for the fact that the level of difficulty in eight questions was unique.

Studies on the measurement by many researchers have been increased since the late 1980s when the development of multidimensional measurement tools started. However, the examination at a point tends to change, depending on age, subject, and situation, for the reason that researchers present grounds, on the basis of various statistical techniques. The importance of validity is emphasized because the integrated perspective of validity is highlighted, and the level of validity tends to be verified as a single concept in recent years [16]. In this study, the validity of the taekwondo electronic protector cognition scale was verified through item response theory as a new method of inspection theory from a new concept of validity perspective based upon the data collected.

The main results from this background implied the following—the electronic protector is the equipment with functional characteristics where the scoring intensity is according to the weight category, and it is set to strike a scoring part of each other with various techniques; the score is obtained by measuring automatically. This electronic protector system was introduced in taekwondo competitions and was applied until now for the purpose of fairness and scientific documentation as a countermeasure for several problems, such as boring matches involving athletes and biased referee decisions, since the proper intensity of hitting power was acknowledged as a score in the system. However, along with these positive outcomes, there was also a drawback, i.e., players who have a substantial match as main agents in the competition had to adjust to the game changes without having a say about athletic performance.

Firstly, the category of five-points response in the taekwondo electronic protector cognition scale was reviewed as appropriate from the examination result. The reason for reviewing the validity of response categories was that the scale reliability becomes low as the respondents get confused if there are too many categories, while there is also a drawback response in the respondents if it cannot be discerned if the categories are too scarce [13]. Therefore, it is meaningful that the response categories of the taekwondo electronic protector cognition scale similar to the one in this study can obtain a proper and reliable response result in order to verify whether it is appropriate for the infit and outfit [23]. As a result of exploring an optimum response category, the validity of the response category standard was appropriate as all of the categories, i.e., categories 1, 2, 3, 4, and 5, were within the range of 0.75 to 1.30. In addition, it was also confirmed that the category of the five-points scale number was appropriate as the coefficient increased step by step as the standard number increased in Andrich’s Threshold.

Secondly, it was confirmed that all the items appeared to be appropriate from the result of item validity and item difficulty in the taekwondo electronic protector cognition scale. The item was reported as inadequate in case the respondents recognized the corresponding item as ambiguous, or the corresponding item had nothing to do with potential variables that were being measured [13]. In other words, this research showed that there were no items that were too easy to respond to for the respondents with high private properties, or too difficult to agree with for the respondents with low private properties, in the Taekwondo electronic protector cognition scale. However, if the logit value (which refers to the item difficulty) was investigated, item 7 (It is easy to obtain the score when it is accurately aligned with the sensor than when it is just strongly stroked), item 3 (Although the kick intensity is weak, the kick accurately aligned with the sensor is recognized as the score), item 1 (Inaccurate attack accidentally tipped on the sensor is connected with the score), and item 18 (Even correct offense is not connected with the score in many cases) were considered as difficult to respond. On the contrary, this research indicated that item 6 (There is no interest in the game because the scoring intensity is too strong.), item 14 (It is crude in appearance), item 24 (I do not know the correct scoring parts), and item 9 (The frequency of kick decreases as it is easy to obtain a fist score) were easy to respond to. The level of difficulty in psychological response means that the corresponding reaction reflects or accommodates reality [24]. The respondents of this study showed that the drawback of the electronic protector in a taekwondo match was that it could not effectively discern the problems in a competitive situation. The development of the electronic protector in taekwondo and the effort of its application were initiated due to the never-ending problems of fairness and referee decisions in the match. In order to ensure the fairness of the referee’s decision, the electronic protector was developed and applied to the field with a discussion that the result could no longer solely rely upon referee due to the mechanical measuring device in which the scoring intensity was judged and suggested by the machine [3,25]. Although there are positive aspects, such as more accurate measurement of the techniques, the fairness of the referee’s decision, and sports modernization [24], there are also various negatives, such as the inconvenience of kick technique motion due to the size and weight, i.e., wearing sensation, sensor response, score intensity, and scoring part; in other words, the problem that the score is connected with those errors was recognized as the biggest one of the electronic protector by many players. Finally, many players pointed out the reason why the predetermined intensity was not connected with a score when they strike the opponent by a kick or fist was originated from sensor malfunction in a situation that there were frequent errors that the sensor was not aware of for some reason. They pointed out that they had the right to doubt the validity of the equipment while they were also aware of the problem of reliability because of the inconsistency of the collected data. However, there was a need to maximize the benefits of athletes who had a match as main agents in the competition, including the technical changes, by using the electronic protector as the positive and negative aspects were presented earlier. In this regard, Kim [18] analyzed that the definite resolution of several phenomena in sports does not appear to recognize and analyze the sports phenomenon itself, while it has to precede looking for the dynamics behind the phenomenon that makes it possible to manifest itself.

Therefore, if the research is provided on the basis of the user experience of players as a subject in a taekwondo match, then that exercise ability is measured as more trustworthy and appropriate, when they have a match with wearing the electronic protector and the result is provided in the sports field. This research aimed at confirming whether the electronic protector cognition scale is appropriate from the perspective of a new concept of validity. In conclusion, the subject reliability and the scale reliability were reported to be appropriate as the degree of validity of infit and outfit was 1.00~1.01. Secondly, the five-points scale was reviewed as appropriate, judging from the phase index in the validity of the standard category number. Third, issues were found in a total of eight items in the analysis of item validity and item difficulty. Finally, the validity of the scale was appropriate, judging from the ability distribution of players and the difficulty of the scale itself.

## 5. Conclusions

This research was conducted with the purpose of presenting future tasks for electronic protector as a measuring tool that can have universal validity as a tool to identify problems with electronic protection by comparing opinions of domestic and foreign taekwondo athletes who participate in the competition wearing ‘progress of electronic protector’ developed to ensure fairness in taekwondo games.

As a result of this study, firstly, the subjects (players) and scales were reliable for the four factors of the strength, fist score, fit, sensor response, and scoring area of taekwondo players on the “Electronic protector Recognition Scale”. Second, the “Taekwondo Electronic Protector Recognition Scale”, consisting of four factors (strength, fist score, wear, sensor reaction), of taekwondo athletes showed that the five-point Likert scale was reasonable as a result of the Andrich threshold analysis. Third, the conformity of questions in 24 paragraphs of the electronic arc recognition scale was appropriate, but as a result of the difficulty evaluation determined by the logit score, it was found that the level of difficulty recognized by the players was unique in eight paragraphs. First score factor’s item 7 (It is easy to obtain the score when it is accurately aligned with the sensor than when it just strongly stroked), intensity factor’s item 3 (Although the kick intensity is weak, the kick accurately aligned with the sensor is recognized as the score) and item 1 (Inaccurate attack accidentally tipped on the sensor is connected with the score), and sensor response and scoring factor’s item 18 (Even correct offense is not connected with the score in many cases) were difficult factors for the players to response, whereas the four factors, namely, intensity factor’s item 6 (There is no interest in the game because the scoring intensity is too strong), wearing sensation factor’s item 14 (It is crude in appearance), sensor response and scoring factor’s item 24 (I do not keno the correct scoring parts), and first score factor’s item 9 (The frequency of kick decreases as it is easy to obtain the first score), were easy factors. In conclusion, it was found that the recognition scale of taekwondo electronic fodder was reasonable in terms of the athletes’ ability distribution and item fit, except for the difficulty level problem in the eight questions.

## Figures and Tables

**Figure 1 ijerph-17-03684-f001:**
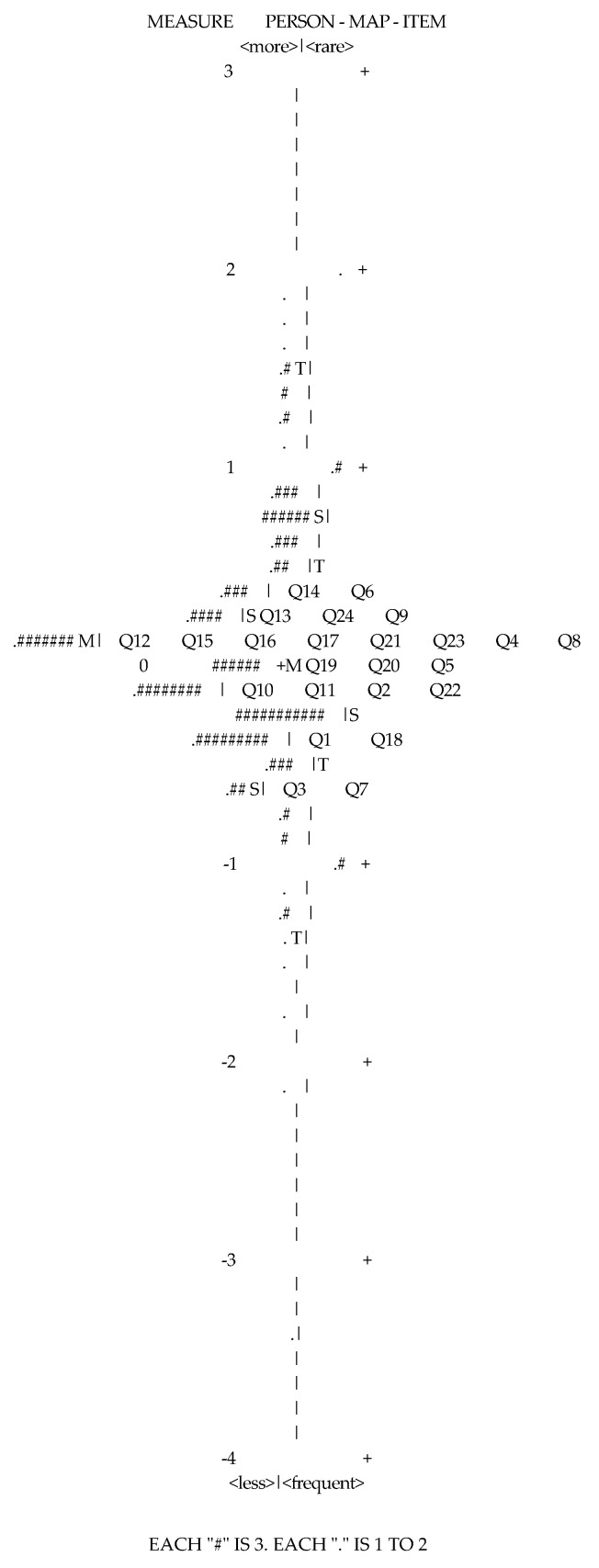
Item goodness-of-fit of the electric protector cognition scale.

**Table 1 ijerph-17-03684-t001:** The distribution of respondents by each continent.

Division	Continent Name	Country Name	*n* (%)
Asia	Asia + Oceania	New Zealand	6 (2.7)
		Philippines	10 (4.4)
		Australia	7 (3.1)
		Korea	130 (57.5)
		4 Countries	153 (67.7)
America	South America + North America	Brazil	2 (0.9)
		Cuba	2 (0.9)
		USA	12 (5.3)
		Canada	3 (1.3)
		4 Countries	19 (8.4)
Europe	-	Norway	15 (6.6)
		Denmark	4 (1.8)
		Sweden	7 (3.1)
		Swiss	5 (2.2)
		Spain	3 (1.3)
		UK	6 (2.7)
		Austria	1 (0.4)
		Jordan	2 (0.9)
		Italy	1 (0.4)
		Croatia	7 (3.1)
		France	2 (0.9)
		11 Countries	54 (23.9)
	5 Continents	19 Countries	226 (100)

**Table 2 ijerph-17-03684-t002:** The characteristics of respondents (male and female) by the continent according to the weight category.

Weight Category	Male (*n*, (%))	Female (*n*, (%))
Fin	14 (12.8)	17 (14.5)
Fly	10 (9.2)	21 (17.9)
Bantam	9 (8.3)	14 (12.0)
Feather	15 (13.8)	16 (13.7)
Light	13 (11.9)	15 (12.8)
Welter	17 (15.6)	12 (10.3)
Middle	10 (9.2)	14 (12.0)
Heavy	14 (12.8)	7 (6.0)
Total	102 (93.6)	116 (99.1)
Missing value	7 (6.4)	1 (0.9)
226(100.0)	109 (48.2)	117 (51.8)

**Table 3 ijerph-17-03684-t003:** The components of the questionnaire.

Division	Research and Variable		Number of Items
The Electronic Protector Cognition	Intensity	1, 2, 3, 4, 5, 6	6
	Fist score	7, 8, 9, 10, 11	5
	Wearing sensation	12, 13, 14, 15, 16	5
	Sensor response and scoring part	17, 18, 19, 20, 21, 22, 23, 24	8
Total items			24 items

**Table 4 ijerph-17-03684-t004:** Subject reliability of taekwondo electronic protector cognition scale.

	TOTAL SCORE	COUNT	MEASURE	MODEL ERROR	INFIT		OUTFIT	
					MNSQ	ZSTD	MNSQ	ZSTD
MEAN	74.0	24.0	0.08	0.22	1.01	−2.0	1.01	−0.2
S.D.	14.1	0.0	0.71	0.03	0.55	2.0	0.55	2.0
MAX.	113.0	24.0	2.59	0.58	3.28	5.6	3.30	5.7
MIN.	27.0	24.0	−3.39	0.21	0.19	−4.8	.20	−4.7
REAL RMSE 0.25 TRUE SD 0.66 SEPARATION 2.67 PERSON RELIABILITY 0.88								
MODEL RMSE 0.22 TRUE SD 0.67 SEPARATION 3.00 PERSON RELIABILITY 0.90								
S. E. OF PERSON MEAN = 0.044								

MNSQ: N show standardized fit statistics, ZSTD: Standardized as a Z-score, S.D: standard deviation, MAX: maximum, MIN: minimum, S.E: standard error, RMSE: root mean squared error.

**Table 5 ijerph-17-03684-t005:** Item reliability of taekwondo electronic protector cognition scale (Measured ITEM).

	TOTAL SCORE	COUNT	MEASURE	MODEL ERROR	INFIT		OUTFIT	
					MNSQ	ZSTD	MNSQ	ZSTD
MEAN	820.7	266.0	0.0	0.07	1.00	−2.0	1.01	0.0
S. D.	59.8	0.0	0.26	0.0	0.19	2.3	0.19	2.3
MAX.	962.0	266.0	0.41	0.07	1.47	5.3	1.54	5.8
MIN.	725.0	266.0	−0.62	0.07	0.70	−4.2	0.71	−3.9
REAL RMSE 0.07 TRUE SD 0.25 SEPARATION 3.68 PERSON RELIABILITY 0.93 MODEL RMSE 0.07 TRUE SD 0.25 SEPARATION 3.82 PERSON RELIABILITY 0.94 S. E. OF PERSON MEAN = 0.05								

S.D: standard deviation, MAX: maximum, MIN: minimum, S.E: standard error, RMSE: root mean squared error.

**Table 6 ijerph-17-03684-t006:** Category validity of taekwondo electronic cognition scale.

CATEGORY LABEL	CATEGORY SCORE	OBSERVED COUNT, %	OBSVD AVRGE	SAMPLE EXPECT	INFIT	OUTFIT	ANDRICH THRESHOLD	CATEGORY MEASURE
					MNSQ	MNSQ		
1	1	584, 9	−0.79	−0.70	0.92	0.94	NONE	(−2.66)
2	2	1317, 21	−0.29	−0.29	0.95	0.96	−1.28	−1.14
3	3	2175, 34	0.08	0.03	1.02	1.07	−0.63	−0.03
4	4	1587, 25	0.37	0.39	1.01	1.01	0.52	1.12
5	5	721, 11	0.80	0.85	1.08	1.09	1.40	(2.73)

**Table 7 ijerph-17-03684-t007:** Item goodness-of-fit and item difficulty.

ITEM	TOTAL SCORE	TOTAL COUNT	MEASURE	MODEL S. E.	INFIT		OUTFIT		PT−MEASURE		EXACT	MATCH
					MNSQ	ZSTD	MNSQ	ZSTD	CORR.	EXP.	OBS%	EXP%
Q1	916	266	−0.41	0.07	0.98	0.2	1.07	0.9	0.38	0.52	38.0	40.3
Q2	860	266	−0.17	0.07	1.11	1.4	1.12	1.5	0.43	0.53	38.0	40.4
Q3	953	266	−0.58	0.07	1.26	2.9	1.29	3.2	0.26	0.51	40.6	40.4
Q4	791	266	0.13	0.07	0.96	−0.5	0.99	−0.1	0.52	0.53	39.5	39.4
Q5	811	266	0.04	0.07	0.89	−1.4	0.97	−0.3	0.41	0.53	42.1	39.7
Q6	725	266	0.41	0.07	0.94	−0.8	0.93	−0.9	0.65	0.53	38.7	38.6
Q7	962	266	−0.62	0.07	1.31	3.5	1.29	3.3	0.28	0.51	35.0	40.6
Q8	806	266	0.07	0.07	1.00	0.1	1.02	0.2	0.50	0.53	38.3	39.6
Q9	770	266	0.22	0.07	0.95	−0.6	0.94	−0.7	0.61	0.53	42.5	39.0
Q10	846	266	−0.11	0.07	0.93	−0.9	0.92	−0.9	0.54	0.53	40.25	40.2
Q11	838	266	−0.07	0.07	91	−1.2	0.90	−1.2	0.60	0.53	42.5	40.1
Q12	782	266	0.17	0.07	0.80	−2.8	0.80	−2.6	0.67	0.53	42.1	39.2
Q13	777	266	0.19	0.07	0.83	−2.2	0.85	−2.0	0.64	0.53	46.6	39.0
Q14	746	266	0.32	0.07	0.97	−0.4	0.97	−0.3	0.60	0.53	41.4	38.8
Q15	799	266	0.09	0.07	1.41	5.3	1.54	5.8	0.32	0.53	35.7	39.5
Q16	781	266	0.17	0.07	0.91	−1.1	0.90	−1.3	0.66	0.53	41.0	39.2
Q17	785	266	0.15	0.07	1.17	2.0	1.16	2.0	0.48	0.53	39.1	39.2
Q18	903	266	−0.35	0.07	1.03	0.4	1.04	0.6	0.54	0.52	36.5	40.2
Q19	829	266	−0.03	0.07	0.74	−3.5	0.76	−3.2	0.57	0.53	46.6	40.0
Q20	810	266	0.05	0.07	1.14	1.8	1.15	1.8	0.49	0.53	34.6	39.7
Q21	793	266	0.12	0.07	0.98	−0.2	0.98	−0.2	0.62	0.53	40.6	39.4
Q22	844	266	−0.10	0.07	0.71	−4.1	0.71	−3.9	0.65	0.53	48.1	40.2
Q23	801	266	0.09	0.07	0.70	−4.2	0.72	−3.9	0.59	0.53	46.6	39.6
Q24	768	266	0.323	0.07	1.21	2.6	1.20	2.4	0.62	0.53	34.6	39.0
MEAN	820.7	266.0	0	0.07	1.00	−0.2	1.01	0			40.4	39.6
S. D.	59.8	0	0.26	0	0.19	2.3	0.19	2.3			3.8	0.6

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
