# Peer review of "The Psychometric Characteristic of the Taekwondo Electronic Protector Cognition Scale: The Application of the Rasch Model"

_ijerph, 2020, doi:10.3390/ijerph17103684_

Round 1

Reviewer 1 Report

The Psychometric Characteristic of the Taekowndo Electronic Protector Cognition Scale: The Application of the Rash Model

IJERPH (ISSN 1660-4601)

The objective this study was to investigate the psychometric characteristics of the electronic protector cognition scale by the infit and outfit of Taekwondo athletes.

The approach of the study appears very original. The contents of the manuscript are quite interesting by his methodology and through the tools of quantification used. The low number of subjects did not affect the quality of the manuscript.

The manuscript reads smoothly and is easy to understand.  The aims, scope, and results of the study are clearly stated.  I have very much enjoyed reading this paper. I find it interesting and clearly written, and satisfying also all the other publication criteria of the “IJERPH”. The study provides a very valuable addition to this line of research, and adds relevantly to the subject with additional original findings. I thus find that this paper definitively delivers results that will surely be of interest to the readership of the journal “IJERPH”. I recommend the publication of this paper.

Author Response

Thank you for your interested in our investigation.

 There is a some changes with other reviewer, i hope you like some changes.  (More details)

Reviewer 2 Report

This manuscript entitled “The Psychometric Characteristic of the Taekowndo Electronic Protector Cognition Scale: The Application of the Rash Model” primarily aimed to recognize the psychometric characteristics, applies them to scale propriety, and then investigates whether the infit and outfit is suitable using the inspection theory of item response theory. The authors bring an interesting study, but there are still some problems that can not up this review to a publishing level. Some suggestions are listed in the specific comments below.

Abstract:

Line 18-19, please, delete “item goodness-of-fit”, which was repetitive.

Introduction:

I suggest that you should improve the description at line 30-31 to provide more detail.

Line 56-61, the sentence should be improved to ensure that an international audience can clearly understand your text.

Materials and Methods

Line 104-105, It is recommended to add detail exclusion and inclusion criteria filters.

Line 131 and line 134, please state manufacturer, city and country from where the two kind of software have been sourced.

Results

I suggest that you should add abbreviations and their explanations under the table 4, such as S.D., MAX, MIN.

Discussion

Please highlight the main findings of your study in the first paragraph of the discussion part.

Conclusions

Please, just demonstrate the main findings in this section (being more "conclusively").

Author Response

Thank you for giving me the opportunity to submit a revised draft of my manuscript titled "The Psychometric characteristic of the Taekwondo Electronic Protector Cognition Scale: The Application of the Rasch model". We appreciate the time and effort that you and the reviewers have dedicated to the reviewers for their insightful comments on my paper. We have been able to incorporate changes to reflect most of the suggestions provided by the reviewers. We have highlighted the changes within the manuscript.

Here is a point-by-point response to the reviewers' comments and concerns.

Abstract:

  1. Line 18-19, please, delete “item goodness-of-fit”, which was repetitive.

Author response: Thanks, I did not recognize it, it has been deleted.

Introduction:

  1. I suggest that you should improve the description at line 30-31 to provide more detail.

Author response: There is a more details…

For instance, a full-body swimsuit in a swimming competition plays an important role in the performance by increasing momentum and reducing low thrust, and the body and mind of athletes whose shoes change frequently in a marathon race.

  1. Line 56-61, the sentence should be improved to ensure that an international audience can clearly understand your text.

Author response: Sensor malfunction in electronic arcs, plate arrest due to referee decision errors, sensor failure attached to the torso protector due to continuous competition, problem of set sensor strength according to Taekwondo player weight class, inaccuracy of major foot skills and strikes, incorrect marking of scores recognized for scoring, problem of size of electronic catch, sensor and body protector technology attached to the foot sensor and body protector due to introduction of electronic protector.

Materials and Methods

  1. Line 104-105, It is recommended to add detail exclusion and inclusion criteria filters.

Author response:  The competition was attended by 143 countries, 1,011 athletes, 250 of whom answered the questionnaire, and 226 people used the questionnaire to analyze the data. I was curious about what the players in many countries thought about e-hocs, and I tried my best to answer the questionnaires as much as possible. Therefore, it was important to gather opinions from many players, not set standards, and the results presented included the country and the number of cases. The classification of continental athletes in Table 2 also shows how the gender and weight classes of the athletes in the questionnaire are distributed. In other words, it would be good to see the results as a result of describing the general characteristics of the collected subjects. The weight category indicates that there is a difference between men and women (e.g., male pinweight: -54kg, female pinweight: -47kg).

  1. Line 131 and line 134, please state manufacturer, city and country from where the two kind of software have been sourced.

Author response: Thanks, we has been changed as following…  

IBM SPSS STATISTICS version 23 (IBM SPSS, Inc., Chicago, IL, USA)

WINSTEPS version 3. 92 (SWREG, Inc. Shannon, Clare, IRELAND)

Results

  1. I suggest that you should add abbreviations and their explanations under the table 4, such as S.D., MAX, MIN.

Author response: Thanks, we has been added legend under the table 4

Table Legends

Table 4. Subject Reliability of Taekwondo Electronic Protector Cognition Scale.

S.D. : Standard Deviation

MAX. : Maximum

MIN. : Minimum

S.E. : Standard Error

RMSE. : Root Mean Squared Error

Discussion

  1. Please highlight the main findings of your study in the first paragraph of the discussion part.

Author response: Thanks, we has been added main findings of our study

The main points found in this study were first, the results of Taekwondo players' response to four factors: strength, fist score, fit, sensor response, and score point for "electronic fixtures" showed that the reliability of the athletes themselves and confidence in the electronic fodder recognition scale were reliable. Second, the 5 Likert scale of the Taekwondo Electronic Hogu recognition scale, which consists of four factors (strength, fist score, wear, sensor reaction), was reasonable according to the Andrich Treshold analysis. Third, as a result of the analysis of the conformity of questions in 24 questions of the electronic arc recognition scale, there were no problems within the statistical range of suitability, but as a result of the difficulty evaluation determined by the logit score, it was found that the level of difficulty recognized by the players was unique in 8 questions. In conclusion, the recognition scale of Taekwondo's e-hogu was found to be a reasonable measure in the ability distribution and question-and-answer drawings of athletes, except for the fact that the level of difficulty in eight questions was unique.

Conclusions

  1. Please, just demonstrate the main findings in this section (being more "conclusively").

Author response: it was changed to…As a result of this study, firstly, the subjects (players) and scales are reliable for the four factors of the strength, fist score, fit, sensor response and scoring area of Taekwondo players on the "Electronic Lodge Recognition Scale". Second, the "Taekwon Electronic Hogu Recognition Scale" consisting of four factors (strength, fist score, wear, sensor reaction) of Taekwondo athletes showed that the 5 Likert scale was reasonable as a result of the Andrich Treshold analysis. Third, the conformity of questions in 24 paragraphs of the electronic arc recognition scale was appropriate, but as a result of the difficulty evaluation determined by logit score, it was found that the level of difficulty recognized by the players was unique in 8 paragraphs. First score factor’s item 7 (it ie easy to obtain the score when it is accurately aligned with the sensor than when it just strongly stroked), intensity factor’s item 3 (Although the kick intensity is weak, the kick accurately aligned with the sensor is recognized as the score) and litem 1 (inaccurate attack accidentally tipped on the sensor is connected with the score), and Sensor Response and Scoring factor’s item 18 (Even correct offence is not connected with the score in many cases) was a difficult factor for the players to response, whereas the four factors intensity factor’s item 6 (There is no interest in the game because the scoring intensity is too strong.), Wearing Sensation factor’s item 14 (It is crude in appearance), Sensor Response and Scoring factor’s item 24 (I do not keno the correct scoring parts,), and First score factor’s item 9 (The frequency of kick decreases as it is easy to obtain a first score)) was an easy factor. In conclusion, it was found that the recognition scale of taekwondo electronic fodder was reasonable in terms of the athletes’ ability distribution and item fit, except for the difficulty level problem in the five questions.
